# Dispersion state phase diagram of citrate-coated metallic nanoparticles in saline solutions

Sebastian Franco-Ulloa [1], Giuseppina Tatulli[2], Sigbjørn Løland Bore [3], Mauro Moglianetti [2], Pier Paolo Pompa[2,4 ✉], Michele Cascella [3,4 ✉] & Marco De Vivo [1,4 ✉]

The fundamental interactions underlying citrate-mediated chemical stability of metal nanoparticles, and their surface characteristics dictating particle dispersion/aggregation in aqueous solutions, are largely unclear. Here, we developed a theoretical model to estimate the stoichiometry of small, charged ligands (like citrate) chemisorbed onto spherical metallic nanoparticles and coupled it with atomistic molecular dynamics simulations to define the uncovered solvent-accessible surface area of the nanoparticle. Then, we integrated coarse-grained molecular dynamics simulations and two-body free energy calculations to define dispersion state phase diagrams for charged metal nanoparticles in a range of medium's ionic strength, a known trigger for aggregation. Ultraviolet-visible spectroscopy experiments of citrate-capped nanocolloids validated our predictions and extended our results to nanoparticles up to 35 nm. Altogether, our results disclose a complex interplay between the particle size, its surface charge density, and the ionic strength of the medium, which ultimately clarifies how these variables impact colloidal stability.

[1] Molecular Modeling and Drug Discovery Lab, Istituto Italiano di Tecnologia, via Morego 30, 16163 Genoa, Italy. [2] Nanobiointeractions & Nanodiagnostics, Istituto Italiano di Tecnologia, via Morego 30, 16163 Genoa, Italy. [3] Department of Chemistry and Hylleraas Centre for Quantum Molecular Sciences, University of Oslo, P.O. Box 1033 Blindern, 0315 Oslo, Norway. [4] These authors jointly supervised this work: Pier Paolo Pompa, Michele Cascella, Marco De Vivo. ✉email: pierpaolo.pompa@iit.it; michele.cascella@kjemi.uio.no; marco.devivo@iit.it

Metal nanoparticles (NPs) with different composition, morphology, and surface chemistry can be used for applications like NP-mediated catalysis[1–4], cancer therapy[5–7], and chemosensing[8–11]. To ensure the solubility of pristine metallic NPs in polar solvents, stabilizing agents like citrate and tetraoctylammonium bromide must be introduced into the mixture. This is exemplified by the Turkevich wet synthetic method, in which metallic NPs are obtained via the reduction of metal-containing chlorine acids in the presence of sodium citrate[12]. Here an excess of citrate anions acts as a stabilizing agent that jackets newly formed metallic nucleation sites and keeps them from crystal growth, agglomeration, and precipitation[13,14]. However, the chemisorption of citrate onto the assembled metallic surfaces critically depends on variables such as the particle size, the surface charge density, and the ionic strength of the medium in which the NPs are dispersed. These variables are crucial in modulating the physicochemical properties of the resulting NPs (e.g., chemical reactivity and surface plasmon resonance frequency)[15]. Nonetheless, the interplay and relationships of these variables are poorly understood at the molecular level, especially in relation to the dispersion state, which is central when developing new NP-based technologies[16,17].

So far, binding of citrate onto metals has been scrutinized for gold NPs. Specifically, recent studies have shed light on the binding mode and energetics of citrate molecules onto different gold facets[18]. These studies used a wide range of complementary techniques, including density functional theory (DFT) calculations[19], scanning tunneling microscopy[20], and X-ray photoelectron spectroscopy (XPS)[21]. Furthermore, experimental investigations have examined the composition of citrate adlayers sitting over diverse gold surfaces[5,20–23]. For example, Lin et al.[5,22,23] reported a lower threshold of 0.8 bound citrate molecules per nm[2] on gold (111) surfaces, as determined by scanning probe microscopy. Similarly, Park et al.[21] estimated an average coverage of 1.7 citrate molecules per nm[2] on the same gold facets using infrared spectroscopy (IR) and XPS. Other studies have reported larger citrate surface coverages. For example, Rostek and co-workers[24] found a value of 3.1 molecules per nm[2] using elemental analysis of 17-nm-sized gold NPs, and Dominguez and co-workers[25] found a value of 4.7 molecules per nm[2] using XPS on 5-nm-sized gold NPs. While highly informative, these studies did not resolve the relation between the surface citrate density and the charge of the coated surfaces. In particular, these results differ in regard to the equilibrium surface density of citrate on gold. This also leaves unresolved the fundamental question of the effective charge of citrate-capped metallic colloids and how this reflects into the NP dispersion state in solution.

Importantly, the dispersion state of metal NPs in saline solutions determines the interfacial area of the metallic surfaces, which in turns allows essential chemical processes for material science and pharmaceutical applications[26]. The dispersion of NPs has been extensively investigated with experimental and theoretical approaches. From an experimental standpoint, the colloidal stability is often associated with the experimentally measurable ζ-potential, that is, the electrostatic potential at the shear plane of a charged body[27]. Nonetheless, the ζ-potential is a descriptor that accounts only for the electrostatic interactions that NPs exert over one another, yet it disregards the equally important van der Waals forces[28]. From a theoretical standpoint, the colloidal stability (i.e., the inter-particle interaction) is typically modeled by a Yukawa potential, in accordance with the Derjaguin–Landau–Verwey–Overbeek (DLVO) theory. This theory has proven useful in studying processes, such as microbial adhesion[29], polymer association[30], and clay aggregation[31]. However, DLVO lacks an atomistic description of the electrical double layer at the surface of NPs, which precludes the examination of the molecular properties of the mobile electrolytes dissolved in the solvent, and eventually aggregating onto the NP[32,33].

Here we investigate the interplay between NP charge and size and the ionic strength of the electrolytic medium, in relation to the stability of citrate-capped metallic colloids. For this purpose, we first develop a new theoretical framework that determines the surface coverage of charged ligands onto spherical NPs. This model is used to determine the number of citrate molecules bound to gold NPs. In parallel, we generate models for enhanced sampling molecular dynamics (MD) simulations. Free energy calculations, together with ultraviolet–visible (UV-vis) spectroscopy experiments, allowed us to rationalize charged NP dispersion and aggregation in terms of simpler two-body interactions and dissect the driving forces leading to these distinct states. In the end, we combine the results from our theoretical model with our general phase diagrams to describe at the molecular level the dispersion state of citrate-capped gold nanocolloids, a system of paramount importance in nanotechnology.

## Results

**Stoichiometry of citrate chemisorbed onto metal NPs**. The surface coverage of citrate onto gold NPs fluctuates across the current literature, with experimental values ranging from 0.8[22] to 4.7[25] molecules per nm[2] and computational estimates ranging from 0.4[34] to 2.0[35] molecules per nm[2]. Thus, as a first step, we developed a theoretical model to estimate the exact stoichiometry ratio and distribution of the charged ligands forming the protecting adlayer in metallic NPs. Moreover, this approach provided a framework consistent with the models for charged NPs that we would use in the rest of the study.

The theoretical model is based on the thermodynamic cycle shown in Fig. 1a (see "Methods" for details and Supplementary Fig. 1), which computes the free energy of $N$ molecules binding to NPs in solution ($\Delta G_{cap}$). Our model decomposes the binding in solution of $N$ ligands, $\Delta G_{cap}$, into three complementary processes that enclose the thermodynamic cycle, namely, (i) the desolvation of the spherical core and $N$ ligands $\Delta G_{desolv}$, (ii) the binding in vacuum of the ligands onto the core $\Delta G_{bind}$, and (iii) the solvation of the protected NP $\Delta G_{solv}$. Importantly, this framework models the protected NP as a spherical, hydrophobic core with a homogeneous charge distribution that describes the electrostatic mean-field effect of the coating ligands. This approach further simplifies the calculation when separating the solvation energy, $\Delta G_{solv}$, into an apolar $\Delta G_{solv}^{apolar}$ and a polar $\Delta G_{solv}^{polar}$ component. In detail, the polar contribution is calculated with a mean-field formula derived from Newtonian mechanics, whereas the apolar component cancels out with the desolvation energy of the reacting core as contained in $\Delta G_{desolv}$. In this way, our theoretical model requires only two parameters that are: the desolvation energy of the ligand (used to compute the rest of $\Delta G_{desolv}$) and the binding energy of one ligand onto the metallic surface $\Delta E_{bind}$ (Fig. 1a). This formulation offers a general and transferable framework to calculate the surface density of small, charged ligands bound to spherical, rigid cores.

We then used the developed model to determine the ligand density of citrate onto gold NPs. For this, we computed $\Delta G_{cap}$ for spherical NPs of diameter 3.0 nm. The first input, the desolvation energy of one citrate molecule, was derived from computer simulations at a coarse-grained (CG) resolution (Fig. 1b). Specifically, we developed an explicit CG model for citrate at its fully deprotonated form, that is, the most populated state at neutral pH. The explicit CG model for citrate was parametrized against atomistic simulations of citrate in water (see Supplementary Discussion 1 for details). Notably, the protonation state of

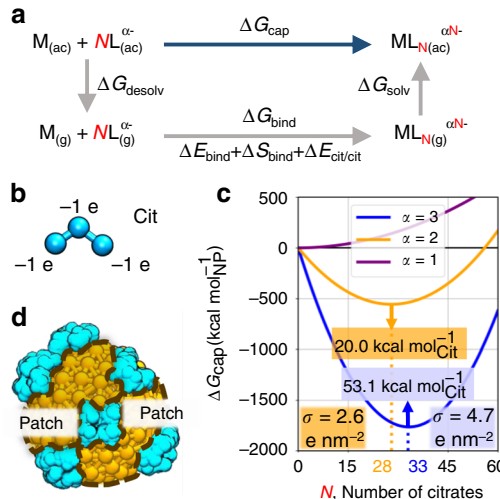

**Fig. 1 Theoretical model for calculating the number of charged ligands bound to a spherical NP. a** Thermodynamic cycle on which the proposed theoretical model is based. $\Delta G_{cap}$ is obtained in this thermodynamic cycle as $\Delta G_{desolv} + \Delta G_{bind} + \Delta G_{solv}$, where $\Delta G_{desolv}$ is the desolvation energy of the NP core (M) and $N$ ligand molecules (L); $\Delta G_{bind}$ is the energy for binding those ligands onto the NP in vacuum; $\Delta G_{solv}$ is the solvation free energy of the resulting capped complex, in which the ligands are modeled as an enveloping uniformly charged sphere. **b** Explicit coarse-grained model for citrate molecule (Cit), which was parametrized against atomistic MD simulations (see Supplementary Discussion 1f). **c** Binding energy as a function of $N$. The model is solved for the three deprotonation states of citrate, i.e., $\alpha = 1, 2, 3$. The energy minima predict the most likely values for $N$ that, in turn, set the surface charge density ($\sigma$) of the NP. **d** Conceptual illustration of a citrate-capped (cyan) nanoparticle showing the presence of catalytically available hydrophobic patches (brown) as suggested by the estimated surface area coverage.

ligands may change when these bind to the NP surface[20]. The second input, the binding energy between citrate and gold surfaces, was taken from DFT-based calculations reported elsewhere[19].

For polyprotic ligands like citrate, the total charge of the capped NP depends on the mean deprotonation state of the bound molecules ($\alpha$). Thus this model can account for single ($\alpha = 1$), double ($\alpha = 2$), or full deprotonation ($\alpha = 3$) of bound citrate molecules, with the latter two being the most populated states at a pH > 4[36], a range favorable for metallic NPs synthesis, extended colloid half-life, and biological assays[37–40]. In this way, the states $\alpha = 2$ and $\alpha = 3$ demarcate a range for the stoichiometric ratio of citrate, which thereby confines the NP surface charge density ($\sigma$). This is in line with experimental evidence on the varying charges of chemisorbed citrate molecules, which enable the formation of H-bond networks at the surface of gold facets[21]. Our model also grasps the pH dependence on the stability of NPs. Particularly, when $\alpha = 1$, the global minimum of $\Delta G_{cap}$ disappears, indicating that the number of ligands that can bind to the NP is not enough to make it soluble in water (Fig. 1c). This is in agreement with the experimentally observed agglomeration of gold NPs at pH ~ 4, conditions at which dihydrogen citrate becomes the dominating species[41].

Using this approach, we found that the computed number of chemisorbed citrate molecules onto the NPs falls into an interval between $N = 28$ (1.31 molecules per nm$^2$, $\sigma = 2.6$ e nm$^{-2}$) and $N = 33$ (1.55 molecules per nm$^2$, $\sigma = 4.7$ e nm$^{-2}$) for the double and full deprotonation of citrate, respectively (Fig. 1c). Given our estimate of citrate molecules chemisorbed onto the NP surface, the net charge of our NPs is expected to range from $-56$ e

($\sigma \sim 2.6$ e nm$^{-2}$) to $-99$ e ($\sigma \sim 4.7$ e nm$^{-2}$), see "Methods" section. Moreover, for each deprotonation state, we also obtained a different value for the binding affinity of citrate onto gold surfaces, which we find in the range of 20.0 ($\alpha = 2$) and 53.1 ($\alpha = 3$) kcal mol$^{-1}$. This range comprises the estimate derived from quantum mechanical simulations in vacuum, i.e., 40.9 kcal mol$^{-1}$[19], supporting the validity of our theoretical framework. In fact, an interpolation of our data at $\alpha = 2$ and $\alpha = 3$ suggests that, to reach a binding affinity of 40.9 kcal mol$^{-1}$, the mean deprotonation state is $\alpha \sim 2.6$, in qualitative agreement with XPS experiments[20].

Importantly, our computed range lies within the boundary values of the experimental measurements of $N \sim 17$ (0.8 molecules per nm$^2$) from Lin et al.[5,22,23] and $N \sim 100$ (4.7 molecules per nm$^2$) from Dominguez et al.[25], which were based on scanning tunneling microscopy and XPS, respectively[42]. In particular, our computed values also match very well the estimate of $N \sim 33$ (1.55 molecules per nm$^2$) from Park and Shumaker-Parry[21], calculated by means of IR and XPS. In addition, Chong and Hernandez[35] performed atomistic MD simulations on citrate molecules explicitly interacting with gold surfaces, predicting $N \sim 42$ (1.98 molecules per nm$^2$), which is also in good agreement with our calculations. Notably, these results agree in spite of additional factors that may affect the binding affinity of citrate for the NPs, including (i) the curvature of the metallic surfaces; (ii) the ratio between metallic (gold) atoms embedded in the bulk, surfaces, and edges; (iii) the steric hindrance between citrate molecules, and (iv) the different proportions of lattice planes.

Our approach also captures the interactions between neighboring citrate molecules and their effect on the polarity of the resulting coated NP. As the citrate molecules become more strongly charged ($\alpha$ increases), the electrostatic repulsion exerted between vicinal ligands encumbers binding onto the NP. In fact, the repulsive energy between citrate molecules $\Delta E_{lig/lig}$ scales as $N^2$, according to our model (see Supplementary Discussion 1 and Supplementary Figs. 1 and 2 for details). In contrast, strongly charged citrate molecules increase the (negative) net charge of the coated NP, thus favoring its dispersion in highly polar solvents like water. This follows from the quadratic relation between the energy from transferring a coated NP between media $\Delta G_{solv}^{polar}$ and the number of bound citrate molecules $N$ (see Supplementary Discussion 1 and Supplementary Figs. 1 and 2 for details).

Another important aspect is the extent to which the binding of citrate onto NPs hinders specific regions of the metallic surface from contacting the solvent. Thus, based on the shape and dimensions of our NP, we first estimated the total surface area of the NP as $4\pi R_{NP}^2$. Then the expected area occupied by a single superficial ligand was computed via all-atoms MD simulations of citrate in water (see Supplementary Discussion 2 for details). We computed the projected area of the van der Waals surface for an individual citrate molecule onto an arbitrary plane. Notably, this calculation is made invariant with respect to the chosen plane, as various orientations are sampled for each MD-generated conformer. We considered 100 orientations for each of the 1000 MD-generated conformers and traced the area's probability density (Supplementary Fig. 3) and the area per citrate of maximum probability (0.16 nm$^2$). In this way, the computed area per citrate, together with the calculated interval of $28 < N < 33$ (see above), returns an estimated surface coverage by citrate molecules in the range between 22.5 and 26.5% for 3.0 nm NPs (Fig. 1d). This range is in good agreement with the lower limit of 35% citrate surface coverage, experimentally estimated for ~40 nm gold NPs[21,41]. This means that ~70–80% of the surface area of such soluble citrate-coated NPs remains in direct contact with the solvent, fully available, for example, for surface catalysis[43–46].

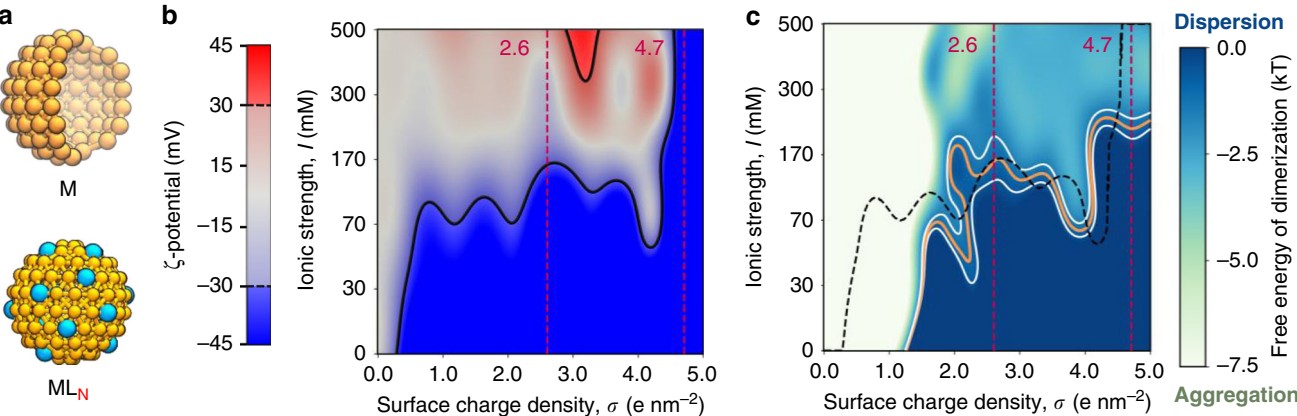

**Fig. 2 Dispersion state phase diagrams of ion-capped metallic NPs. a** Coarse-grained models of pristine (M, top) and capped ($ML_N$, bottom) NPs for the construction of the phase diagrams. The cyan beads implicitly account for grafted ligands with a charge of $-2$ e. **b** The relation between the NP surface charge density ($\sigma$), the environment ionic strength ($I$), and the computed $\zeta$-potential. The black contour demarcates the region where the $\zeta$-potential lies between $-30$ and $+30$ mV. The opaque blue and red regions indicate the conditions where the $\zeta$-potential computations suggest colloidal stability. **c** Map of the NPs free energy of dimerization for the various studied systems. The orange and white curves outline the region where the free energy is $-1.0 \pm 0.5$ kT. The black contour drawn for the $\zeta$-potential is superimposed onto this plot. The dark blue and light green colors indicate the conditions at which the free energy calculations suggest colloidal stability and aggregation, respectively. The dashed, red lines indicate the limiting values of $\sigma$ for citrate-capped NPs as determined by the developed theoretical model.

**Effect of ionic strength and NP charge on the $\zeta$-potential**. To study the aggregation state of citrate-capped NPs, we initially examined the effect of the surface charge density ($\sigma$) of the NPs and the ionic strength of the medium ($I$) on the $\zeta$-potential of nanospheres of 3.0 nm size. For this task, we employed implicit-ligand models of coated NPs (Fig. 2a). In the implemented model, selected surface beads were assigned a charge of $-2$ e to mimic the presence of monohydrogen citrate. The charge per bead chosen thus describes citrate molecules at their most abundant ionization state on capped gold NPs[20], but it extends to hard anionic species like cyclic oxocarbons and dicarboxylic acids. Implicit-citrate models have gained increasing attention in recent years as they reduce the phase space's dimensionality, while offering a reliable representation of systems otherwise too intricate to simulate. These models rely on the covalent character of the gold–citrate interaction. DFT calculations have quantified a binding affinity of 40.9 kcal mol$^{-1}$ [19]. In contrast to chelation and multipolar interactions, the complexation of citrate onto gold surfaces implies the formation of stiff chemical bonds that damps ion competition and ion pairing. Recent XPS experiments have also ratified a weak coupling between bound citrate molecules and sodium counterions present in electrolytic solutions[20]. Implicit-citrate models have already accomplished semi-quantitative agreement when studying processes like membrane rupture[47], protein adsorption[48,49], NP-induced protein denaturation[23], and synchronized NP internalization[50].

By means of CG-MD simulations, we estimated the $\zeta$-potential of the NPs for all possible combinations of $\sigma$ and $I$, following a protocol introduced elsewhere and described in detail in the "Methods" section[51,52]. The $\zeta$-potential of all the systems is mapped into a bidimensional plot displayed in Fig. 2b. The computed value for the $\zeta$-potential ranges between ~$-150$ and $+40$ mV. The shaded region in Fig. 2b encloses the values for $\sigma$ and $I$ at which the $\zeta$-potential of the NPs lies between $-30$ and $+30$ mV, a minimum requirement for colloidal stability[53]. Thus the black contour delimits values of the surface charge $\sigma$ and salt concentration $I$ that separate colloidal stability vs. instability, with the latter reflecting aggregation in experiments. Notably, the $\zeta$-potential is computed as the radial electrostatic potential in the position of the shear plane (see "Methods" for details). The fast decay of the electrostatic potential (described by a Yukawa

potential in Debye–Hückel's theory) produces a sensible response between the position of the shear plane and the $\zeta$-potential, which results in a wider range of admissible values of $\zeta$-potential than those that are experimentally relevant[27,28].

In detail, when the net charge of the NP is $< -13$ e in magnitude ($\sigma < {\sim}0.6$ e nm$^{-2}$), the dispersion state of the NPs shows a linear correlation between the critical ionic strength $I$ and $\sigma$ (Fig. 2b). Then the critical value of $I$ stays in a plateau at ~70 mM for intermediate charges limited by $-13$ e and $-87$ e (i.e., $\sim0.6 < \sigma < \sim4.1$ e nm$^{-2}$). In this interval, as expected, we observed an increased attraction of sodium counterions toward the NP, as $\sigma$ increases. However, we also found that the magnitude of the attraction is such that the increased charge of the NP is quickly screened out by the first adlayer of sodium counterions, which are located within the hydrodynamic radius (i.e., the Stern layer). This means that the measured $\zeta$-potential of these NPs is similar to that of a less charged NP at $I = 0$ ($\sigma < \sim0.6$ e nm$^{-2}$), a situation in which NPs tend to aggregate more easily. This explains the plateau in the dispersion state phase diagram in Fig. 2b. Finally, for NPs with charges $> -87$ e ($\sigma > \sim4.1$ e nm$^{-2}$), the system also acquires very high values of $\zeta$-potentials (~$-100$ mV). In these conditions, the sodium ions are no longer able to fully neutralize the NP charge. This drastic increase in $\zeta$-potential leads to high NP stability, which is also favored by the reduced capacity of the sodium ions in solution to counterbalance the very high charge of the metallic surface. This may be explained by the entropic cost of bringing additional (sufficient) sodium ions onto the highly charged metallic surface (sterically hindered at this point because of sodium saturation).

Based on our results, the electrostatic interactions between a charged NP and an electrolytic solvent fall within one of the regimes described above, according to the net charge of the sphere. These regimes can be classified as (i) depolarized ($\sigma < \sim0.6$ e nm$^{-2}$), where the thermal motion of the sodium counterions overcomes the electrostatic attraction toward the NP; (ii) mildly polarized ($\sim0.6 < \sigma < \sim4.1$ e nm$^{-2}$), in which the Coulomb forces attract enough ions into the Stern layer to screen out the charge load of the NP; and (iii) hyperpolarized ($\sigma > \sim4.1$ e nm$^{-2}$), the situation at which the counterions' steric and electrostatic hindrance, as well as their loss of translational degrees of freedom, limit their binding onto the NP.

**Role of inter-particle interaction for NP dimerization.** The $\zeta$-potential of NPs is a measure of the charge density at the electrophoretic radius of charged bodies, and it is often associated with the stability of colloids. However, this measure does not account for van der Waals forces[28]. Thus, to fully account for inter-particle interactions in colloidal coalescence/aggregation processes, we calculated the free energy of dimerization for our 3.0 nm NPs. We considered all the surface charge densities $\sigma$ and salt concentrations $I$ discussed above and listed in the "Methods" section. Moreover, to describe the dimerization process, we defined a collective variable, CV1, as the minimum distance between the van der Waals surfaces of two interacting NPs. By computing the free energy along CV1 for the dimerization process for all the explored $\sigma$ and $I$, we obtained a dispersion state phase diagram.

Based on the assumption that NP dimerization initiates aggregation, we thus used the region where the free energy is $-1.0 \pm 0.5$ kT (i.e., the minimal free energy for dimerization) to limit the conditions of colloidal (in)stability for metallic suspensions (Fig. 2c). In this way, this region demarcates the conditions at which the systems have a 37% ($1/e$) chance of being dispersed, namely when the thermal motion along CV1 is enough for the NPs to escape the energy well associated with dimerization. Interestingly, this region matches well with the region characterized by a $\zeta$-potential of $\pm 30$ mV, as outlined in Fig. 2b, confirming that electrostatics is a key player for colloidal stability.

Figure 2c suggests that low surface charge densities ($\sigma < \sim 1.2$ nm$^{-2}$) always lead to an aggregated state. A regime with a similar behavior was derived from our $\zeta$-potential calculations ($\sigma < \sim 0.6$ e nm$^{-2}$). It is interesting to note that the threshold is shifted toward higher values of $\sigma$ when the interactions between NPs are considered explicitly. Arguably, this may be due to the large hydrophobic matching between the metallic cores that leans the system toward aggregation. Then, for surface charge densities in the interval of $\sim 1.2 < \sigma < \sim 4.2$ e nm$^{-2}$, our free energy calculations suggest that the system remains stably dispersed under salt concentrations of $\sim 70$ mM. Again, this agrees with the $\zeta$-potential calculations. Similarly, in line with the $\zeta$-potential plot, the free energies of dimerization indicate a change in behavior when $\sigma > \sim 4.2$ e nm$^{-2}$. At these $\sigma$ values, the $\zeta$-potential plot suggests that NPs become more resistant to aggregation. Nonetheless, our free energy calculations allow highly charged NPs to aggregate at $I \sim 170$ mM. Remarkably, over the entire range of $\sigma$, the $I$ for inducing aggregation increases irregularly with $\sigma$, denoting a non-trivial relationship between the critical ion concentration of the solution and the dispersion state of the NPs.

NP aggregation can be induced by increasing the salt concentration of the medium, $I$, in otherwise dispersed systems, as largely reported in experiments[54–56] and also confirmed in our simulations. In this regard, we found that alterations in the aggregation state of the NPs arise mostly from the screening of the NP charge by the sodium counterions in solution. These ions can rest at the interface of the metallic bodies and form salt bridges that stabilize the NP–NP dimer. Our simulations indicate that, as $\sigma$ increases, a halo-like cloud of accumulated counterions is more clearly formed near the contact site of the two approaching NPs (Fig. 3a). To further corroborate this observation, we computed the number density of sodium ions, up to 2.0 nm away from the NPs, with respect to the angle defined by CV1 (Fig. 3b). Notably, this way of analyzing the motion of the counterions in solution provides an explicit and atomic-detailed description of the ion anisotropic placement around the NP dimer. For example, when two NPs of net charge $-40$ e ($\sigma = 1.9$ e nm$^{-2}$) reach each other in a solution of ionic strength $I = 30$ mM, the angular number density of sodium ions near their contact

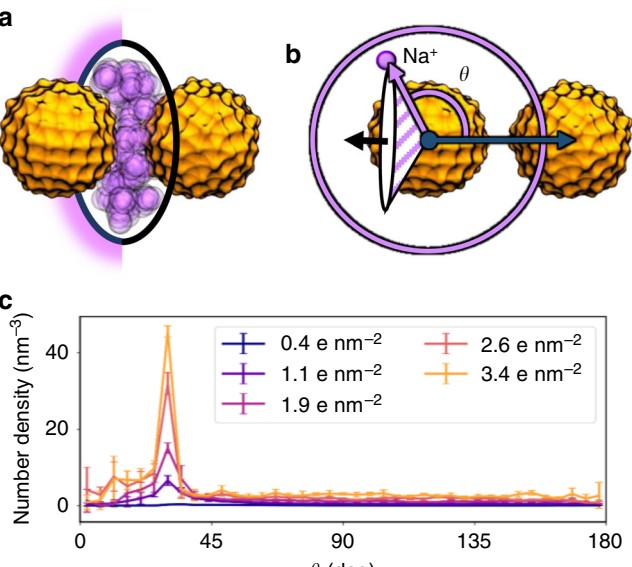

**Fig. 3 Counterion distribution upon NP dimerization. a** Halo-like cloud of counterions (purple) accumulated at the edge of the two NPs contacting region. Superposition of various frames of the simulations. NPs are colored orange. **b** The angle $\theta$ is formed between the direction connecting the two NPs (CV1) and the position vector from one NP center of mass to a sodium ion. Each value of $\theta$ defines a conic volume around the reference NP. **c** The number density of ions as a function of the polar angle $\theta$. The height of the peak found at $\sim 32°$ increases with the surface charge density of the NPs. Error bars indicate the standard deviation calculated over trajectory frames.

region increases to up to 15.0 nm$^{-3}$, as compared to the 1.1 nm$^{-3}$ measured in the opposite end of the contacting axis (Fig. 3c). Similar trends are observed for different salt concentrations. Also, this analysis overcomes limitations of the mean-field-founded DLVO theory at short inter-NP distances[57].

It is important to note that, in the Martini CG force field, the sodium and chloride ions' beads comprise the first solvation shell of the two ionic species. As such, these models offer a qualitative description of the ion mobility around the charged NPs rather than structural insights on their respective binding mode. Nonetheless, we performed supplementary atomistic MD simulations of 3.0 nm citrate-capped gold NPs to corroborate the results from our CG simulations. In these simulations, we computed the time for which sodium counterions remained in the vicinity of the capped NPs. Indeed, in both our CG and atomistic simulations, longer residence times for sodium counterions were found as the surface charge of the NP increased (Supplementary Fig. 5).

In addition, we investigated the merging of the electrical double layers formed around the NPs upon dimerization. This chemical event was described by simulations at different values of CV1 (Fig. 4a). Each of our simulations restrained the system with an additional harmonic potential centered at a specific value of CV1, ranging between 0.0 and 3.0 nm. During these simulations, we monitored the charge density of the overall system and analyzed it for fixed planes perpendicular to CV1.

In total, we focused on four different planes located at: (i) the center of mass of one of the NPs, (ii) half-radius away from an NP's center of mass, (iii) an NP's van der Waals surface, and (iv) the midplane between the two NPs. Specifically, Fig. 4b shows the sliced planes from simulations of NPs with a charge of $-40$ e ($\sigma = 1.9$ e nm$^{-2}$) in an environment of ionic strength $I = 70$ mM. Initially, these simulations describe the states in which CV1 = 0.9 nm (intermediate distance, Fig. 4b top), revealing halos of

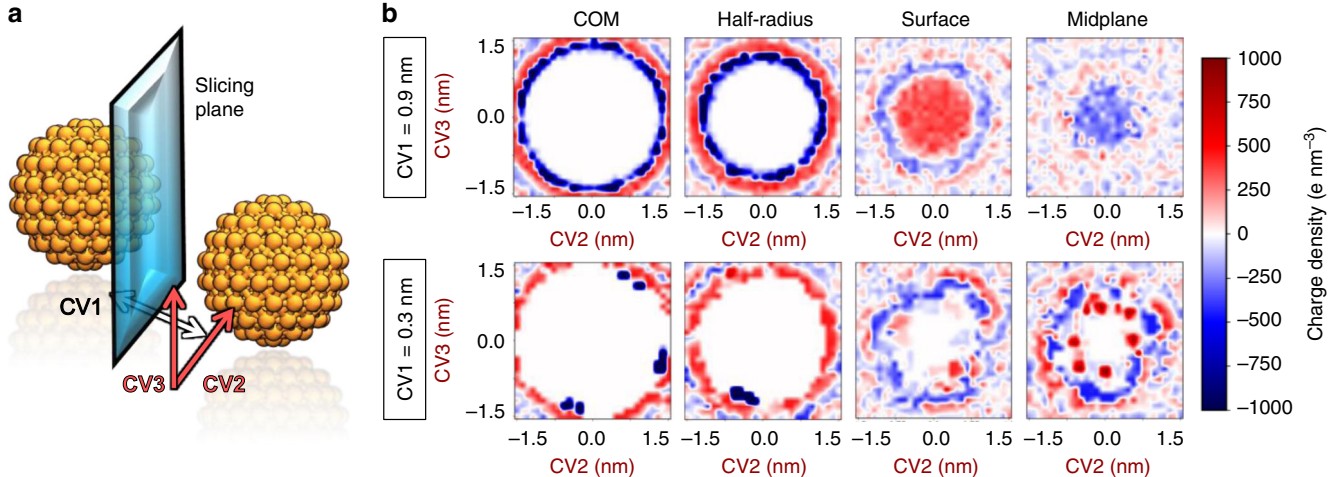

**Fig. 4 Transversal charge density as a function of the distance between the NPs. a** Graphical definition of the collective variables of interest. CV1 is the minimum distance between the two NP surfaces. CV2 and CV3 are two bases perpendicular to CV1 chosen arbitrarily. The position of the slicing plane indicates the cuts on which the systems' charge density was estimated. **b** The charge density of the overall systems averaged over time for various slicing planes at CV1 = 0.9 nm (top) and CV1 = 0.3 nm (bottom). The four studied planes lie at (i) an NP's center of mass (COM), (ii) half-way along the radius of an NP, (iii) at the van der Waals surface of an NP, and (iv) at the midplane between the two NPs. All plots are derived from systems with NPs loaded with a net charge of −40 e ($\sigma$ = 1.9 e nm$^{-2}$) in an environment of ionic strength 70 mM.

alternating charge density sign around the NPs. These halos indicate the presence of a nearly unperturbed electrical double layer. This means that, at CV1 = 0.9 nm, the ion clouds are mildly affected by the presence of the NPs, yet the layers can be outlined (Fig. 4b top), with the lamellar nature of the electrical double layers preserved. As expected, at this value of CV1, the charge of the ligand-representing beads is distributed in a nearly circular shape (Fig. 4b top left corner) due to the free rotation of the NPs around their center of mass during the simulations.

In contrast, at CV1 = 0.3 nm, the sodium counterions bind to specific sites at the surface of the NPs, generating a few regions of high positive charge density at the dimer's interface (Fig. 4b bottom right corner). This result reflects the decreased diffusivity of the sodium ions close to the NPs in solution. Also, the negatively charged beads in the metallic bodies (i.e., the ligand beads) appear in discrete locations, which suggests a restricted exploration of the rotation of the two NPs around their respective center of mass (see Fig. 4b bottom left corner, Supplementary Fig. 4, and Supplementary Discussion 3 for details). That is, the free rotation of the two NPs is sensibly diminished compared to that of the two NPs at longer CV1 values. These results illustrate how the electrical double layers change during dimerization, assisting the strategic accommodation of sodium counterions and the formation of salt bridges for optimal dimer stabilization.

**Linking dimerization of NPs with colloidal stability.** To demonstrate that the multi-body process of NP agglomeration can be rationalized in terms of NP dimerization, we synthesized 3.5, 13.0, and 36.9 nm citrate-capped gold NPs (AuNPs) and we experimentally determined their dispersion state at various salt concentrations. In detail, we tracked the plasmon absorption bands of the synthesized NPs as a function of increasing concentrations of sodium chloride, I (in the 0–120 mM range), and assessed particle aggregation through the aggregates to monomer band ratio (Abs$_{650}$/Abs$_{max}$). In all cases, increasing the salt concentration beyond a critical threshold led to aggregation, as indicated by a step-like shift in absorbance (Fig. 5a and Supplementary Fig. 6). Importantly, the 3.5 nm AuNPs aggregated at a salt concentration of ~90 mM, which is in close agreement with our computed phase diagrams. The 13.0 and 36.9-nm colloids

aggregated at lower ionic strengths, ~60 and 50 mM, respectively. Hence, as the AuNPs become larger, the critical ionic strength for aggregation decreases. This may be explained by the increase in the hydrophobicity of NPs of larger size. Due to the stronger attractive van der Waals term, the NPs would thus require fewer salt bridges (lower values of I) to stabilize the agglomerate.

To test this hypothesis about the effect of increased hydrophobicity of NPs with larger size and to evaluate the energetics associated with aggregation upon NP size enlargement, we performed additional simulations, modulating the forces mutually exerted by the nano-sized spheres. Due to the apolar nature of the gold–gold bonds and the homogeneity of the electron cloud of the metallic bulk, the interaction between an NP and its surroundings are accounted for with a Lennard-Jones potential that models dispersive van der Waals forces. Hence, the van der Waals radius and well depth ($\varepsilon$) dictate the affinity between the NPs and their surroundings, i.e., through their hydrophobic interaction. Specifically, the metal-to-metal Lennard-Jones term was modified from its original value of $\varepsilon$ = 1.35 kT to more attractive potentials, which are contemplated in the standard Martini force field ($\varepsilon$ = 1.75 kT and $\varepsilon$ = 2.10 kT). Computing the free energy along CV1 for NPs with these hydrophobicities, we verified the known linear correlation between the free energy of dimerization and $\varepsilon$ ($R^2$ = 0.99, Supplementary Fig. 7). From this regression, we concluded that, for $\varepsilon$ = 1.41 and 1.47 kT, the free energy of dimerization decreased by ~1.0 and 2.0 kT, respectively.

Using these new models of larger (more hydrophobic) NPs, we computed the corresponding dispersion state phase diagrams (Fig. 5b, c) consistently with those in the previous cases, reported above. Notably, the increased size also influences the curvature of the NP surface, a parameter whose effect on NP aggregation cannot be excluded. Decreasing $\sigma$, or increasing I, leads to values at which the thermal energy cannot prevent aggregation. Moreover, as the hydrophobicity of the NPs becomes stronger (higher values of $\varepsilon$), fewer sodium counterions are needed to bridge the charged surfaces. Consequently, the critical ionic strength I to induce dimerization is shifted toward lower values. In detail, when $\varepsilon$ = 1.41 kT, the critical I lies at ~60 mM, whereas for $\varepsilon$ = 1.47 kT the threshold is further reduced to ~30 mM. This agrees with our experimental measurements, as the computed

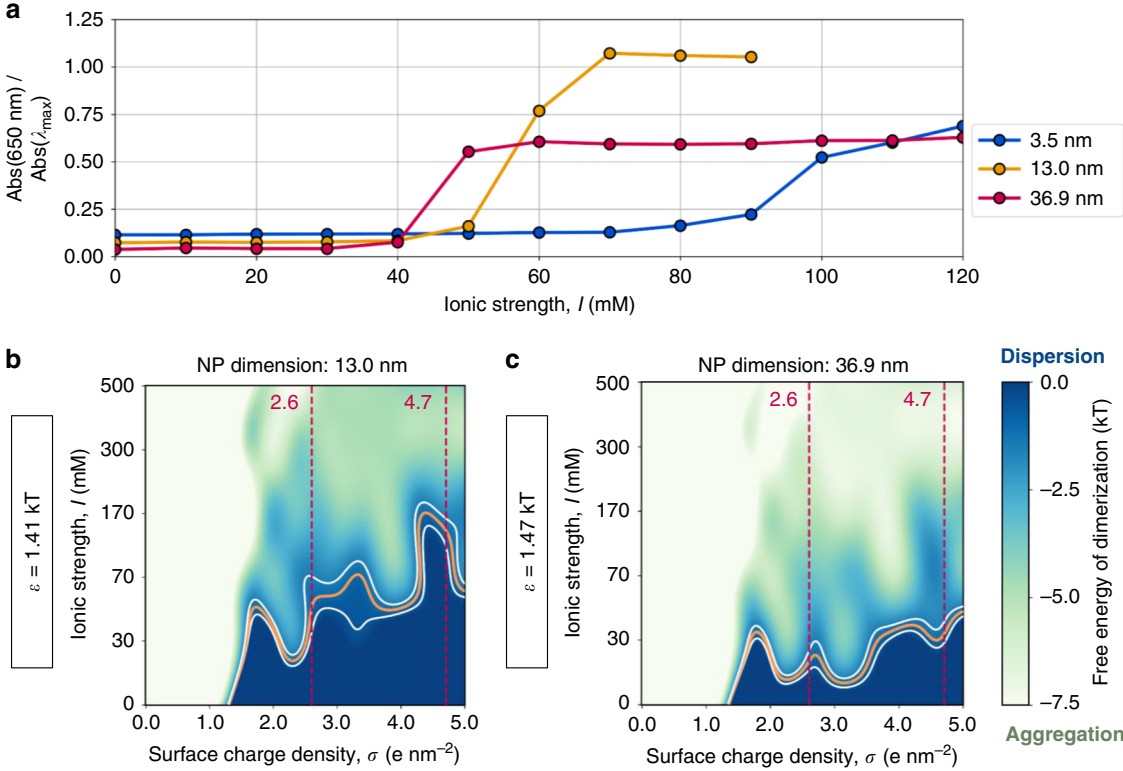

**Fig. 5 UV-vis experiments and colloidal stability at different NP sizes. a** Experimentally measured absorbance of 3.5, 13.0, and 36.9 nm citrate-capped AuNPs at 650 nm relative to the maximum absorbance. The lower panel shows the dispersion state phase diagrams for citrate-capped NPs with varying hydrophobicity. The free energy of dimerization was calculated for our model NPs, modifying the well depth of the van der Waals interaction between metallic beads. The profiles obtained using $\varepsilon = 1.41$ kT (**b**) and 1.47 kT (**c**) are in good agreement with experimental data for 13.0 and 36.9 nm, respectively. The orange and white curves outline the regions where the free energy is $-1.0 \pm 0.5$ kT. The dark blue and light green indicate the conditions at which the free energy calculations suggest colloidal stability and aggregation, respectively. The dashed, red lines indicate the limiting values of $\sigma$ for citrate-capped NPs as determined by the developed theoretical model.

data indicate that, as citrate-capped NPs grow in size, the NP charge needs to be screened out to a lesser extent in order to induce ion-mediated aggregation. These results show that the ion-assisted interactions that govern the dimerization of our modeled 3.0 nm NPs are also valid for larger systems. Hence, these results provide a framework for dispersion state phase diagrams for citrate-capped NPs of varying sizes.

In summary, we examined the aqueous stability of citrate-capped metallic NPs using both computation and experiments. First, a new theoretical model was implemented to determine the citrate coverage and surface charge density of metallic NPs. The computed ligand density was found to be in excellent alignment with experimental data, and it allowed us to characterize the surface properties of citrate-capped gold NP, specifically the charge density and surface coverage. The calculation of the particles' charge density later enabled us to make predictions on their dispersion state based on their $\zeta$-potential and free energy of dimerization. In parallel, by studying NPs with different surface charge values in saline solutions of various ionic strengths, we unraveled the driving forces that lead to nanocolloid aggregation. Our $\zeta$-potential calculations indicated that the critical ionic strength for inducing NP aggregation varies non-linearly with the surface charge density of the spherical NPs. Consequently, depending on the surface charge density, the interaction pattern between the capped NPs and an electrolytic medium may be categorized as depolarized, mildly polarized, or hyperpolarized.

To fully account for dispersion forces between multiple NPs during coalescence, we also estimated the free energy of NP dimerization using CG-MD simulations. The dispersion state phase diagram derived from these calculations is in good agreement with our $\zeta$-potential map, further supporting the critical role of electrostatics in NP aggregation. Moreover, our analyses of the sodium counterions' motion in solution unveiled the formation of halo-like structures of sodium, which promotes ion-assisted NP dimerization through salt bridges. Finally, UV-vis absorbance experiments validated our free-energy-based dispersion state phase diagram and allowed us to extend our observations to NPs of up to 35 nm. In this regard, our results indicate that, as the NP hydrophobicity increases, the concentration of ions required to induce aggregation is reduced.

Altogether, our results are a step toward rationalizing the complex relationship between the particle size, the surface charge density, and the ionic strength of the medium, offering new fundamental insights into the mechanism by which these variables modulate colloidal stability.

## Methods

**Development of the theoretical model.** The free energy required to cap with ligands a metallic NP, $\Delta G_{cap}$ ($N$), was partitioned into three different components according to the thermodynamic cycle exhibited in Fig. 1a. In this figure, the horizontal processes point to the complexation of the ligand onto an NP, whereas the vertical ones point to the (de)solvation of the various parties. Thus, by minimizing the change of free energy $\Delta G_{cap}$ we recover the value of $N$ of the highest likelihood. The first contribution, $\Delta G_{desolv}$, can be expressed in terms of the desolvation energy of the metallic NP ($\Delta G_{desolv}^{M}$) and that of the ligand ($\Delta G_{desolv}^{lig}$), as shown in Eq. (1) (Supplementary Fig. 1a). The first of these terms cancels out with the apolar contribution of $\Delta G_{solv}$ (see below), and the second was calculated by

means of CG-MD through the thermodynamic integration approach (see Supplementary Discussion 1a for computational details).

$$\Delta G_{\text{desolv}}(N) = \Delta G_{\text{desolv}}^{\text{M}} + N\Delta G_{\text{desolv}}^{\text{lig}}. \quad (1)$$

The second contribution, $\Delta G_{\text{bind}}$, corresponds to the binding of $N$ molecules onto the metallic NP (Supplementary Fig. 1b). This contribution comprises three terms: (i) the electronic binding energy of the ligand onto the metallic (gold) surface ($\Delta E_{\text{bind}}$) obtained from high-level quantum calculations discussed elsewhere[19], (ii) the entropy of binding ($\Delta S_{\text{bind}}$) estimated from the classical theory of ideal gases (found to be negligible), and (iii) a correction to account for the repulsion between multiple chemisorbed ligands ($\Delta E_{\text{lig/lig}}$) derived from the classical definition of electrostatic potential energy. The generic form of $\Delta G_{\text{bind}}$ is shown in Eq. (2) and discussed in detail in Supplementary Discussion 1b, c.

$$\Delta G_{\text{bind}}(N) = N(\Delta E_{\text{bind}} - T\Delta S_{\text{bind}}) + \Delta E_{\text{lig/lig}}(N). \quad (2)$$

The last term in the thermodynamic cycle, $\Delta G_{\text{solv}}$ (Eq. (3)), accounts for the solvation of the capped complex, and it was divided into an apolar and a polar term (Supplementary Fig. 1c). The apolar term cancels out with $\Delta G_{\text{desolv}}^{\text{M}}$, whereas the polar component arises from the classical interpretation of electrostatic potential energy (see Supplementary Discussion 1d for details).

$$\Delta G_{\text{solv}}(N) = \Delta G_{\text{apolar}}^{\text{ML}_N} + \Delta G_{\text{polar}}^{\text{ML}_N}. \quad (3)$$

By solving this model for the double ($\alpha = 2$) and full deprotonation ($\alpha = 3$) of citrate, we showed that the limiting values for the number of citrate molecules chemisorbed onto our 3.0 nm NPs were $N = 28$ and $N = 33$. These values, in turn, fix a charge range for these NPs between $-56$ and $-99$ e. The lower value $-56$ e is equal to $-2$ e per citrate molecule multiplied by $N = 28$ citrates, i.e., $\sigma = 2.6$ e nm$^{-2}$. The upper value $-99$ e is equal to $-3$ e per citrate molecule multiplied by $N = 33$ citrates, i.e., $\sigma = 4.7$ e nm$^{-2}$.

**NP modeling and $\zeta$-potential calculations.** For our CG-MD simulations, we modeled metallic NPs as hollow spheres made from 126 beads arranged as stacked rings[58–60]. The size of the NPs was measured at the van der Waals surface of the resulting structure (3.0 nm). The beads were assigned the type C1, as implemented in the Martini v2.2 force field, and a mass of 556 u.m.a. to account for the bulk internal beads. The spherical shape of the NPs was retained by imposing an elastic network with a force constant of 15000 kJ mol$^{-1}$ nm$^{-2}$. The elastic network united each bead with its six nearest neighbors as well as their farthest neighbor. In the case of ion-capped NPs, beads were randomly chosen and assigned a partial charge of $-2$ e until the target net charge was reached. Here we explored 14 values for the surface charge density $\sigma$, equidistantly sampled in a range between 0.0 and 5.0 e nm$^{-2}$.

To determine the $\zeta$-potential of our modeled NPs, we performed CG-MD simulations of our NPs in saline solutions. One capped NP was initially placed in a simulation box, leaving a minimum distance to the faces of 3.5 nm. The box was then immersed in water using the refPol force field[61,62]. Additional polarizable sodium and chlorine ions[62] were included to reach ionic strengths of 0, 30, 70, 170, 300, and 500 mM. A minimization was carried out using the steepest descent method to relax the solvent around the particles. The systems were then thermalized to 310 K and pressurized to 1 bar in the span of 1 ns in the NPT ensemble. For this, the V-rescale thermostat ($\tau_T = 2.0$ ps) and the isotropic Berendsen barostat ($\tau_P = 5.0$ ps and $\kappa = 4.5 \times 10^{-4}$ bar$^{-1}$) were activated, and an integration time step of 2 fs was used. For the 100-ns-long production runs, we increased the integration time step to 20 fs and implemented the Parrinello–Rahman barostat ($\tau_P = 14.0$ ps and $\kappa = 4.5 \times 10^{-4}$ bar$^{-1}$)[63]. All bonds were constrained with the LINCS algorithm[64]. Short-range nonbonded interactions were calculated within a radius of 1.2 nm from each bead, whereas long-range electrostatic interactions were considered using the fourth-ordered PME method[65]. All simulations were conducted with Gromacs-2019.2[66].

To calculate the shear plane, we adopted a methodology similar to the one used by Heikkila et al.[52]. The shear plane, the radial position at which the distribution of the sodium counterions starts deviating from Debye–Hückel's description, was derived from the simulations at $I = 0$. We calculated the radial distribution function (RDF) of the sodium ions with respect to the NPs' center of mass. Then the function $Ae^{-Br}/r + C$ was fitted to the RDF curves. The fitting was performed for the points at distances greater than $r_i$, where $r_i$ assumed 20 equidistant values in the range between 1.9 and 3.0 nm. Thus, for each NP (i.e., different $\sigma$ values), 20 fittings were performed. From each of the fittings, the shear plane was defined as the position in which the RDF deviated by at least 0.1 from its fitted curve. Consequently, for each value of $\sigma$, we obtained a distribution of positions (Supplementary Fig. 8). The final shear plane was the median of these distributions. Finally, the reported $\zeta$-potentials correspond to the electrostatic potential (calculated by means of Gauss' Law) at the NPs' corresponding shear plane.

**Free energy calculations.** To calculate the free energy of NP dimerization, we performed potential of mean force calculations. Two capped NPs were initially placed with their center of mass 6.0 nm away from each other (CV1 = 3.0 nm, see Supplementary Fig. 9). Then the systems were solvated, ionized, and minimized as in the $\zeta$-potential calculations. The systems were then thermalized to 310 K and pressurized to 1 bar in the span of 1 ns in the NPT ensemble. For this, the V-rescale thermostat ($\tau_T = 2.0$ ps) and the isotropic Berendsen barostat ($\tau_P = 5.0$ ps and $\kappa = 4.5 \times 10^{-4}$ bar$^{-1}$) were activated, and an integration time step of 2 fs was used. This was followed by a second equilibration under the same conditions, increasing the time step to 10 fs for 5 ns. Once the systems had reached the intended temperature and pressure, the Berendsen barostat was replaced for its Parrinello–Rahman counterpart ($\tau_P = 14.0$ ps and $\kappa = 4.5 \times 10^{-4}$ bar$^{-1}$)[63]. The rest of the parameters were set as in the single-NP simulations discussed above.

To sample the dimerizing reaction coordinate, we performed steered MD simulations in which the two rigid bodies were dragged toward each other along CV1 (the minimum distance between the NP van der Waals surfaces). The equilibrium distance between the center of mass of the NPs was thus reduced at a rate of 0.125 nm ns$^{-1}$ with a force constant of 5000 kJ mol$^{-1}$ nm$^{-2}$. To estimate the free energy of aggregation, we used the umbrella sampling method. When the particles were far away from each other (CV1 > 1.2 nm), windows were extracted every 0.1 nm. The value of CV1 was restrained with a harmonic force constant of 5000 kJ mol$^{-1}$ nm$^{-2}$ [67]. Each of these windows was simulated for 25 ns, and the first 5 ns were discarded as equilibration for the weighted histogram analysis method[68]. When the particles were closer together (CV1 $\leq$ 1.2 nm), windows were extracted every 0.05 nm, and the same force constant (5000 kJ mol$^{-1}$ nm$^{-2}$) was used to restrain the systems. These windows were subject to a simulated annealing in which the systems were heated to 450 K within the first 5 ns and cooled until 310 K in the span of 20 ns. Then production runs of 225 ns were launched. These parameters were found to be optimal for exploring the NPs' rotational degrees of freedom (see Supplementary Discussion 3 for a discussion on the various protocols assessed).

**NP synthesis and absorbance experiments.** The 3.5-nm citrate-capped AuNPs were synthesized as reported in the literature[69]. In a round-bottomed flask, 0.6 mL of freshly prepared 0.1 M NaBH$_4$ (Sigma-Aldrich-Merck) was added to 100 mL of an ice-cold aqueous solution containing 0.25 mM HAuCl$_4$ (Alfa Aesar) and 0.25 mM trisodium citrate (Sigma-Aldrich-Merck), while stirring. The suspension turned red-brown immediately, due to NP formation. For the 13.0-nm citrate-capped AuNPs, we used the classical Turkevich–Frens method[70]. In all, 150 mL of a 0.25 mM aqueous solution of HAuCl$_4$ were transferred to a two-neck round-bottomed flask, connected to a bulb condenser, and placed on a heating mantle. After reaching boiling point, 25 mL of 38.8 mM aqueous solution of trisodium citrate were added. The solution was kept gently boiling for 30 min, until a red wine color appeared, indicating NP formation. Lastly, the 36.9-nm citrate-capped AuNPs were prepared by seeded growth of 15 nm AuNPs[69]. In a round-bottomed flask, under vigorous stirring and at room temperature, 1 mL of 15 nm AuNPs was diluted into 120 mL of MilliQ water, followed by the addition of 0.4 mL of 0.1 M hydroxylamine sulfate (Sigma-Aldrich-Merck) solution. Then 10 mL of 2 mM aqueous solution of HAuCl$_4$ were added dropwise. After that, 2.65 mL of 0.1 M trisodium citrate solution were added to stabilize the NPs.

**Absorbance profiles.** In a low-volume cuvette, 100 µL of 400 pM AuNPs were added to 300 µL of increasing concentrations of an NaCl solution (0 mM–120 mM) in 2 mM trisodium citrate. UV-vis spectra (190–840 nm) were acquired by a NanoDrop 2000c UV-Visible Spectrophotometer (Thermo Fisher Scientific).

**Transmission electron microscopic imaging.** Transmission electron microscopy images of AuNPs deposited on 300 mesh carbon-coated grids were acquired with a TEM JEOL-JEM 1011 microscope (Supplementary Fig. 10).

## Data availability
The three-dimensional models and topology parameter files used throughout this study have been made available at a public repository (https://github.com/cebasfu93/CitrateNanoparticleAggregation). The raw data supporting our findings, as well as MD simulation trajectories, are available from the corresponding authors upon request.

## Code availability
The theoretical model has been implemented with Python-v3.7. The raw MD simulation trajectories discussed in this work have been analyzed, and later plotted, with in-house scripts. All the code has been made available in a public repository (https://github.com/cebasfu93/CitrateNanoparticleAggregation).

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

## Acknowledgements

M.D.V. thanks the Italian Association for Cancer Research (AIRC) for financial support (IG 23679). This work was also supported by the Research Council of Norway through the CoE Hylleraas Centre for Quantum Molecular Sciences (Grant number 262695) and by the Norwegian Supercomputing Program (NOTUR) (Grant number NN4654K).

## Author contributions

S.F.-U. performed all the simulations. S.F.-U., S.L.B., M.C., and M.D.V. have developed the theoretical and computational work. G.T., M.M., and P.P.P. planned and performed the experimental work. All authors analyzed the data. The manuscript was written by contributions from all authors.

## Competing interests

The authors declare no competing interests.
