## [Peer Review File · Nature Communications]

REVIEWER COMMENTS

Reviewer #1 (Remarks to the Author):

In this paper, Franco-Ulloa et al. use theoretical models and all atom and coarse-grained simulations to estimate the stoichiometry of citrate molecules absorbed onto spherical metallic NPs, exploring the effect of this on the NP aggregation in different conditions (ionic strength, etc.).

The work is potentially interesting. However, there are essentially two reasons why I don't believe that this work is very well suited for publication in Nature Communications. First, on a technical level I find some potential limitations in the approaches used here by the authors that make me doubting of the generality of the conclusions that are obtained (see points below). Second, on a scientific level I find the paper very detailed, but at the same time better suited for a more technical computational journal. Overall, I find that this work aims at tackling a specific system in detail, but the general conclusions, or the transferability of such conclusions to other systems or toward general concepts remains unclear.

Some critical points are detailed below:

1. The authors use different approaches for different tasks that, while reading the text, are not always clear. For example, it is confusing what has been done with atomistic and what with coarse grained models, and especially why. Since the beginning of the paper, when presenting the theoretical model for the ΔG calculation in Figure 1 the authors refer the reader to the SI. But a convincing motivation for using this theoretical model, what are the model limitations, the particular (or general) cases in which the adopted model is valid or not are missing. Since the choice of a specific model implies a reason why this has been chosen, why this is well suited for studying the systems (while others for example are not) and also specific limitations in terms of validity of the results that can be obtained, I believe that in this sense the paper does not clearly deliver the message.
2. From Figs. 1b and S8, it seems that the Citrate-capped NP models have been built mimicking a case where the Cit ions are explicitly bound onto the NP surface, while the Cit-capped NP is then immersed in a solution with a certain ionic strength (free ions in water). However, in such a case there is no real competition between the Cit ions on the NP surface and the ions present in solution (no Cit unbinding from the NP surface). Is this reliable? Why? I believe that in the systems the ionic effects can be way more complex than this, and a continuous competition between the ions in the system are always present. The validity of such CG model should be thus proven.
3. Also, in the SI the authors explain that the Cit-capped NP models have been modeled by replacing some beads in the NP surface with other beads having a $-2e$ charge (see Fig 1 and S8) - each bound Cit ion being represented by one single $-2e$, $-1e$ or $-3e$ charged bead. I don't understand this choice. First the authors say that they developed a CG model for Cit composed of three connected $-1e$ charged CG beads (Fig S8b), and then they assume that when one Cit binds to the NP, from the outside the coordinated Cit appears as a single charged bead (where the charge changes with the deprotonation level)? This is confusing. Such an explicit-bonded model is valid only in a case with no ionic competition is present in the system (irreversible Cit binding). Moreover, if I interpret this correctly, I believe that the choice of a $-2 e$ charge (or of a one $-1 e$ charge) on the Cit bead may be related to the assumption that one (or two) of the 3 charges of Cit is involved in the electrostatic binding with Au atoms, and thus this is a lost (neutralized) charge. This to me looks as an oversimplification: in a non-bonded model the Cit molecules bound to the NP surface would keep appearing as three -1 charged beads, no? But even in such a case, the Cit should be able to move on the NP surface to some extent. Moreover, I believe that one single bead with a $-2e$ or $-3e$ total charge is anyways different from 2 or 3 bound beads with -1 charge each, as it is the essence of multivalent interactions.
4. Again related to this bound Cit model: If I am not wrong in Figure S4 the authors quantify the residence time of Na^+ ions on NP in the order of 100-150 ns. First it is not clear with which type of model these residence times have been collected. If with CG models, then the authors should be very careful, as the solvation of the ions can be strongly different in CG models compared to all atom ones (as it would be that of the Cit ions, which nonetheless are coordinated and irreversibly bound to the NP surface).

5. The plot of Fig 1c is unclear to me. From the used model it seems that the ΔG_{cap} is the free energy associated to the binding of the citrates onto the NP surface (coming from the solution). Is this correct? The presence of the minimum makes me hypothesizing that this is a global ΔG – namely, the global free energy gain – and that this is not normalized by the number of ions. Nonetheless, the energies in the plot look enormous in my opinion. At $\alpha=3$, with 33 Cit ions this gives ~ 1700 kcal/mol? This is roughly 51 kcal/mol of binding free energy per-Cit ion. This looks like a very high free energy for the interaction of one Cit ion with the surface of the NP, which indicates a super-strong and irreversible Cit binding and that Cit unbinding is not allowed. Is this realistic? If this has been obtained with the CG models, or via the theoretical model, this should be proven and validated by means of atomistic simulations, to ensure that the Martini-based CG models are accurate in this sense. The solvation of Cit ions can be, for example, strongly different in atomistic and CG models. In this sense, the free energies obtained with CG models can be different from the atomistic ones as the entropy in the models is different.

6. It is unclear how the theoretical model used by the authors treat ionic competitions and cooperative effects. For example, when the ionic strength increases in the system, don't the authors observe ionic clusters in solution and on the surface of NPs? I believe that this can have strong local effects that are poorly captured by average analyses and exceed simple additive approaches. This makes me wondering about the accuracy of the theoretical model used. Moreover, in CG models, where ions are typically modeled as CG beads representing one ion with its solvation shell, such local clustering effects could be impaired.

7. Last, it seems to me that the take home message of this paper aims at being general. The authors clearly say in the final part of the conclusion that their results clarify how the size of the NPs, their surface charge density and the ionic strength control the aggregation of the NP. In my opinion this is true only for the specific system studied herein and in particular for the conditions explored in the models. In fact, if the authors keep all these features constant, but increase or decrease the temperature in the system, the results would be probably different. This is a key reason why I believe that, provided that the technical points mentioned above are clarified, this work in its current form provides an interesting study on a specific case, but I doubt about the general relevance of the study on a wider perspective.

8. Overall, I find the text and the whole paper a bit too technical for a high-impact journal targeting a broad readership, while this would be better suited for a more technical specific journal.

Reviewer #2 (Remarks to the Author):

This article aims to determine the surface coverage of citrates on a gold nanoparticle using a thermodynamic cycle that allows them to treat the effects of solvation on the acquisition of citrates through the free energies of solvation of the nanoparticles with and without citrates, and the free energy of the acquisition of citrates in vacuum. Their results are not without experimental or computational precedent in the literature, but the work adds to the on-going research aimed at resolving this important quantity.

They noted the work of Park et al in providing an estimate of the citrate charge density at 1.7 citrate molecules per nm^2 versus the work by Rostek and co-workers with reported an value of 3.1 molecules per nm^2 . The computational work done here to resolve the correct density has precedent in the work of Chong et al, J. Phys. Chem. 2018, 122, 28393-28401 using a very different approach. That work favored the densities seen by Park et al, and less. They also appear to anticipate the values found here for coverage of the various protonated citrates. The results of the present work should be compared and contrasted accordingly.

In summary, this work is worthy of publication but only once it considers and contrasts its results with all of the literature precedent.

ISTITUTO ITALIANO
DI TECNOLOGIA

Reply to the Reviewers

First, we acknowledge the Reviewers' attentive reading of the manuscript. Below, we report in black a point-by-point response to the Reviewers, whose comments are in red.

Reviewer 1:

In this paper, Franco-Ulloa et al. use theoretical models and all atom and coarse-grained simulations to estimate the stoichiometry of citrate molecules absorbed onto spherical metallic NPs, exploring the effect of this on the NP aggregation in different conditions (ionic strength, etc.). The work is potentially interesting. However, there are essentially two reasons why I don't believe that this work is very well suited for publication in Nature Communications. First, on a technical level I find some potential limitations in the approaches used here by the authors that make me doubting of the generality of the conclusions that are obtained (see points below). Second, on a scientific level I find the paper very detailed, but at the same time better suited for a more technical computational journal. Overall, I find that this works aims at tackling a specific system in detail, but the general conclusions, or the transferability of such conclusions to other systems or toward general concepts remains unclear.

Some critical points are detailed below:

1. The authors use different approaches for different tasks that, while reading the text, are not always clear. For example, it is confusing what has been done with atomistic and what with coarse grained models, and especially why. Since the beginning of the paper, when presenting the theoretical model for the ΔG calculation in Figure 1 the authors refer the reader to the SI. But a convincing motivation for using this theoretical model, what are the model limitations, the particular (or general) cases in which the adopted model is valid or not are missing. Since the choice of a specific model implies a reason why this has been chosen, why this is well suited for studying the systems (while others for example are not) and also specific limitations in terms of validity of the results that can be obtained, I believe that in this sense the paper does not clearly deliver the message.

We thank the Reviewer for describing our work as potentially interesting. Based on the Reviewer's comments, we now discuss more exhaustively the developed theoretical model in the 'Results and Discussion' section. We explain that the implementation of such a model was motivated by the lack of consensus found in the literature in regards to the surface coverage of citrate onto metallic nanoparticles. Specifically, experimental scientists have reported citrate coverages on gold surfaces ranging from 0.8 (Lin, Y., et al. **2003**, *Langmuir* DOI: 10.1021/la0350251) to 4.7 (Bajaj, M., et al. **2020**, *Sci. Rep.* DOI: 10.1038/s41598-020-65013-0) molecules per nm². Similarly, computational studies have employed analogous densities between 0.4 (Tavanti, F., et al. **2015**, *New J. Chem.* DOI: 10.1039/C4NJ01752H) and 2.0 (Chong, G., et al. **2018**, *J. Phys. Chem. C* DOI: 10.1021/acs.jpcc.8b09666) molecules per nm².

Taking into account the variability of the reported coverages, we opted for developing a new theoretical framework consistent with the coarse-grained models of the charged nanoparticles that we would implement in the rest of the study. In the text, our new model is formulated in general terms for small capping agents being conjugated onto rigid, spherical nanoparticles. The developed framework requires only two inputs, i.e. 1) the ligand's binding energy onto the metallic surface in vacuum and 2) the ligand's solvation energy. The first of these values was taken from high-level DFT

ISTITUTO ITALIANO
DI TECNOLOGIA

calculations reported elsewhere (Al-Johani, H., et al. **2017**, *Nat. Chem.* DOI: 10.1038/nchem.2752). For the second term, we used a thermodynamic integration scheme on an explicit coarse-grained model developed for citrate. Please note that the explicit citrate model was here parametrized against all-atoms molecular dynamics simulations of citrate in water. The methodological details of these simulations are described in section S1F.

Based on the comments from this Reviewer, we have realized that the source of confusion lies in the multiple modeling strategies adopted throughout the work. Nonetheless, it is a matter of fact that the aggregation of citrate-capped metallic nanoparticles is governed by a plethora of phenomena occurring at multiple scales. As such, we concluded that only a cohesive combination of computational methods and theoretical models that complement each other's reach and limitations would be able to address such complex chemical process. Hence, in the revised version, we now distinguish more clearly the level-of-theory and the resolution of each of the implemented models. We have added two new short paragraphs in the first subsection of the 'Results and Discussion'. These new paragraphs motivate the formulation of our theoretical model and, notably, discuss in further detail its transferability. Here we report the new paragraphs:

"The theoretical model is based on the thermodynamic cycle shown in Figure 1A (see Methods for details and Figure S1), which computes the free energy of N molecules binding to NPs in solution (ΔG_{cap}). Our model decomposes the binding of N ligands in solution, ΔG_{cap} , into three complementary processes that enclose the thermodynamic cycle, namely, (i) the desolvation of the spherical core and N ligands ΔG_{desolv} , (ii) the binding in vacuum of the ligands onto the core ΔG_{bind} , and (iii) the solvation of the protected nanoparticle ΔG_{solv} . Importantly, this framework models the protected NP as a spherical, hydrophobic core with a homogeneous charge distribution that describes the electrostatic mean-field effect of the coating ligands. This approach further simplifies the calculation when dividing the solvation energy, ΔG_{solv} , into an apolar and a polar component. In detail, the polar contribution is calculated with a mean-field formula derived from Newtonian mechanics, whereas the apolar component cancels out with the desolvation energy of the reacting core as contained in ΔG_{desolv} . In this way, our theoretical model requires only two parameters, that is, the desolvation energy of the ligand (used to compute the rest of ΔG_{desolv}) and the binding energy of one ligand onto the metallic surface ΔE_{bind} (Figure 1A). This formulation offers a general and transferable framework to calculate the surface density of small, charged ligands bound to spherical, rigid cores.

We then used the developed model to determine the ligand density of citrate onto gold nanoparticles. For this, we computed ΔG_{cap} for spherical NPs of diameter 3.0 nm. The first input, the desolvation energy of one citrate molecule, was derived from computer simulations at a coarse-grained resolution. Specifically, we developed an explicit coarse-grained model for citrate at its fully deprotonated form, that is, the most populated state at neutral pH. The explicit coarse-grained model for citrate was parametrized against atomistic simulations of citrate in water (see section S1 for details). Notably, the protonation state of ligands may change when these bind to the NPs surface.²⁸ The second input, the binding energy between citrate and gold surfaces, was taken from high-level DFT calculations reported elsewhere.²⁷"

We also included an additional paragraph in section S1F to clarify the usage of our atomistic simulations.

ISTITUTO ITALIANO
DI TECNOLOGIA

“The explicit-citrate coarse-grained model was parametrized against atomistic MD simulations. In detail, the equilibrium angle θ_0 was obtained from a 100 ns- long, all-atom MD simulations of citrate (parametrized with the GAFF force field18) in TIP3P water.¹⁹ This simulation was also used to calculate the three main moments of inertia for a citrate molecule (i.e., I_i in section S1C). In addition, the same atomistic simulation was used to obtain the projected area per citrate molecule onto our AuNPs (i.e., ‘a’ in section S2).”

2. From Figs. 1b and S8, it seems that the Citrate-capped NP models have been built mimicking a case where the Cit ions are explicitly bound onto the NP surface, while the Cit-capped NP is then immersed in a solution with a certain ionic strength (free ions in water). However, in such a case there is no real competition between the Cit ions on the NP surface and the ions present in solution (no Cit unbinding from the NP surface). Is this reliable? Why? I believe that in the systems the ionic effects can be way more complex than this, and a continuous competition between the ions in the system are always present. The validity of such CG model should be thus proven.

Indeed, the coarse-grained models employed in our study consist of spherical bodies with charged beads on the surface that implicitly account for chemisorbed citrate molecules. This family of models have gained increasing attention in recent years as they reduce the phase space’s dimensionality, while offering a reliable representation of systems otherwise too intricate to simulate. Importantly, the success of these models relies on the strength of the citrate-gold bonds, that is, 40.9 kcal mol⁻¹. This value was calculated from high-level DFT calculations, and it proofs the covalent character of the binding (Al-Johani, H., et al. **2017**, *Nat. Chem.* DOI: 10.1038/nchem.2752). In contrast to chelation and multipolar interactions, the complexation of citrate onto gold surfaces implies the formation of stiff chemical bonds that damps ion competition.

In a similar regard, a recent experimental study performed a thorough characterization of citrate-capped and thiol-functionalized gold nanoparticles (Park, J. W., et al. **2014**, DOI: 10.1021/ja4097384). These XPS experiments indicate that the citrate ions chemisorbed on gold nanoparticles interact only weakly with sodium ions from electrolytic solutions, like the ones considered in our study. Remarkably, the same study also reveals a surprising resistance from citrate molecules to their displacement by other anions (e.g. chloride), even during ligand-exchange reactions.

The experimental and quantum mechanical studies described above support that ion-exchange processes are rare (unfavorable) on the surface chemistry of citrate-capped nanoparticles, sustaining the validity of implicit-citrate models on the basis of a nearly-covalent binding. Notably, these findings have led to extensive literature where intrinsic citrate models are used to study the interaction between metallic surfaces and macromolecules, reaching semi-quantitative agreement with experimental results. Some interesting examples include Lee, K., et al. **2018**, *ACS Nano* DOI: 10.1021/acsnano.8b00759, Li, Y., et al. **2017**, *Nanotechnology* DOI: 10.1088/1361-6528/aa56e0, and Brancolini, et al. **2015**, *ACS Nano* DOI: 10.1021/nn506161ij. These publications report on the effect of citrate-capped gold nanoparticles on protein denaturation, membrane rupture, and liposome

ISTITUTO ITALIANO
DI TECNOLOGIA

formation, effectively reproducing experimental observations and further endorsing the value of implicit-citrate models.

In light of the Reviewers' comment, we consider these points could have been better addressed throughout the manuscript, as they represent critical considerations that sustain the reach of our coarse-grained models. With this in mind, we have included a paragraph in the 'Methods' section that discusses the validity of our models in the context of the gold-citrate chemistry and recent success stories. Here we report the new paragraph:

"For this task, we employed implicit-citrate models for the coated NPs. Within these models, selected surface beads were assigned a charge of $-2e$ that mimics the presence of monohydrogencitrate, that is, the most abundant ionization state on capped gold NPs.²⁸ Implicit-citrate models have gained increasing attention in recent years as they reduce the phase space's dimensionality, while offering a reliable representation of systems otherwise too intricate to simulate. These models rely on the covalent character of the gold-citrate interaction. High-level DFT calculations have quantified a binding affinity of $40.9 \text{ kcal mol}^{-1}$.²⁷ In contrast to chelation and multipolar interactions, the complexation of citrate onto gold surfaces implies the formation of stiff chemical bonds that damps ion competition and ion pairing. Recent XPS experiments have also ratified a weak coupling between bound citrate molecules and sodium counterions present in electrolytic solutions.²⁸ Implicit-citrate models have already reached semi-quantitative agreement when studying processes like membrane rupture,⁶² protein adsorption,^{63,64} NP-induced protein denaturation,³¹ and synchronized NP internalization.⁶⁵

3. Also, in the SI the authors explain that the Cit-capped NP models have been modeled by replacing some beads in the NP surface with other beads having a $-2e$ charge (see Fig 1 and S8) - each bound Cit ion being represented by one single $-2e$, $-1e$ or $-3e$ charged bead. I don't understand this choice. First the authors say that they developed a CG model for Cit composed of three connected $-1e$ charged CG beads (Fig S8b), and then they assume that when one Cit binds to the NP, from the outside the coordinated Cit appears as a single charged bead (where the charge changes with the deprotonation level)? This is confusing. Such an explicit-bonded model is valid only in a case with no ionic competition is present in the system (irreversible Cit binding). Moreover, if I interpret this correctly, I believe that the choice of a $-2 e$ charge (or of a one $-1 e$ charge) on the Cit bead may be related to the assumption that one (or two) of the 3 charges of Cit is involved in the electrostatic binding with Au atoms, and thus this is a lost (neutralized) charge. This to me looks as an oversimplification: in a non-bonded model the Cit molecules bound to the NP surface would keep appearing as three -1 charged beads, no? But even in such a case, the Cit should be able to move on the NP surface to some extent. Moreover, I believe that one single bead with a $-2e$ or $-3e$ total charge is anyways different from 2 or 3 bound beads with -1 charge each, as it is the essence of multivalent interactions.

To clarify this point, it is necessary to distinguish between our three distinct modeling approaches, namely 1) our new theoretical model, 2) our all-atom molecular dynamics simulations, and 3) our coarse-grained molecular dynamics simulations. Each of these strategies provides insights on different yet complementary processes involved during nanoparticle dimerization. Our theoretical model calculates the number of ligands (e.g., citrate) bound to a rigid, spherical core. The model is

ISTITUTO ITALIANO
DI TECNOLOGIA

formulated based on a thermodynamic cycle involving three transformations: i) the reagents' desolvation, ii) the binding of the ligands onto the core in vacuum, and iii) the complex's solvation.

The first of these terms (i) includes the desolvation energy of a single citrate molecule (multiplied by N). To compute this quantity, we developed an explicit coarse-grained model for citrate compatible with the models that we would later use for our simulations on nanoparticles (i.e., based on the Martini force field). This model consisted of three beads with a charge of $-1e$, each. Note that the explicit-citrate coarse-grained model is only used to calculate the citrate's desolvation energy, and we therefore consider the fully deprotonated state, which is known to be the most populated state in solutions at neutral pH. The bonded parameters of the explicit-citrate model were derived from an atomistic simulation of a citrate molecule in water.

Furthermore, the third term in our theoretical model (iii), requires calculating the solvation energy of the coated nanoparticle. For this calculation, the capped NP is modelled as a hydrophobic sphere with a uniformly charged shell around it that accounts for the mean-field electrostatics ascribed to the capping ligands. This modeling strategy allowed us to derive a mean-field formula based on the definition of the classical electrostatic energy that depends only on the total charge of the NP. In turn, the total charge of the NP depends on the number of bound ligands, N , and their mean deprotonation state, α . It is important to note that the deprotonation state of bound citrate molecules is not necessarily the same as in their free, fully-solvated state. In fact, Park and co-workers (Park, J. W., et al. **2014**, *J. Am. Chem. Soc.* DOI: 10.1021/ja4097384) demonstrated experimentally that $\alpha \sim 2$ for citrate-capped gold nanoparticles.

In parallel to the development of the theoretical model, we assessed the effect of surface charge on nanoparticle dimerization. For this, we built three-dimensional models of the nanoparticles, assigning a charge of $-2e$ to a variable number of surface beads. A value of $-2e$ was chosen based on the findings of Park, et al. for citrate, but our results hold for equally charged ligands like cyclic oxocarbons and dicarboxylic acids. Note that the obtained phase diagrams are independent from our theoretical model. In fact, the theoretical model serves as a complement that provides a test case (citrate-capped gold nanoparticles in this case), allowing us to illustrate the reach of the diagrams.

In addition, we required specific properties of citrate in water throughout our study that we calculated using atomistic molecular dynamics simulations. Specifically, we derived three properties from an all-atom simulation of a citrate molecule in water. First, the principal moments of inertia (i.e., I_i in section S1B) that were required by our theoretical model. Second, the equilibrium angle of our explicit-citrate coarse-grained model (i.e., θ_0 in section S1F). Third, the projected area of a single citrate molecule onto a plane (i.e., a in section S2), which allowed us to estimate the solvent-exposed area of the capped NPs.

ISTITUTO ITALIANO
DI TECNOLOGIA

We realize that the combination of modeling strategies was presented in a confusing manner in the original version of the manuscript, and we do apologize for it. In the new version, we have edited the 'Methods' and 'Results and Discussion' sections in accordance with the Reviewer's comment. We now describe in further detail the need and reach of our theoretical model, as well as the atomistic and coarse-grained simulations. We also clarify the distinction between the theoretical model and the dispersion state phase diagrams, two frameworks developed in parallel that, when combined, offer a complete description of the dispersion state of charged metallic nanoparticles.

4. Again related to this bound Cit model: If I am not wrong in Figure S4 the authors quantify the residence time of Na⁺ ions on NP in the order of 100-150 ns. First it is not clear with which type of model these residence times have been collected. If with CG models, then the authors should be very careful, as the solvation of the ions can be strongly different in CG models compared to all atom ones (as it would be that of the Cit ions, which nonetheless are coordinated and irreversibly bound to the NP surface).

The Reviewer is right at pointing out that the coordination of ion-pair complexes could behave differently according to the model's resolution. Since the ions' beads also account for their first solvation shell, the structural characterization of the chelated complexes goes beyond the scope of coarse-grained methods like the ones here employed. In fact, our analysis of the ions' distributions (illustrated in Figure 3, Figure S4, and now Figure S5) provides a qualitative description of the chemical environment upon nanoparticle dimerization. The residence times in the Supplementary Information, together with Figure 3, indicate that there are some regions where the ions become more labile than others.

We understand that the residence time computed with coarse-grained models has no univocal correspondence with the "real time", and we have made this clear in the revised version. Specifically, we have made this interpretation explicit in the 'Results and Discussion' section, and we have clarified the origin of the reported residence times in the Supplementary Information, as requested by the Reviewer.

Importantly, to further clarify the point raised by the Reviewer, we have performed twelve 250 ns-long atomistic molecular dynamics simulations of individual citrate-capped AuNPs (charge density 0.9, 1.9, and 2.9 e nm⁻²) at different ionic strengths (0, 30, 70, and 170 mM). The new simulations confirm that the residence time of sodium counterions is extended as the charge of the nanoparticle increases, in consistency with our coarse-grained models. These new simulations are discussed in detail in section S3 and Figure S5. The correspondence found between our atomistic and coarse-grained simulations further indicates the validity of implicit-citrate coarse-grained models for studying nanoparticle dimerization.

5. The plot of Fig 1c is unclear to me. From the used model it seems that the ΔG_{cap} is the free energy associated to the binding of the citrates onto the NP surface (coming from the solution). Is this correct? The presence of the minimum makes me hypothesizing that this is a global ΔG – namely, the global free energy gain – and that this is not normalized by the number of ions. Nonetheless, the energies in the plot looks

ISTITUTO ITALIANO
DI TECNOLOGIA

enormous in my opinion. At $\alpha=3$, with 33 Cit ions this gives ~ 1700 kcal/mol? This is roughly 51 kcal/mol of binding free energy per-Cit ion. This looks like a very high free energy for the interaction of one Cit ion with the surface of the NP, which indicates a super-strong and irreversible Cit binding and that Cit unbinding is not allowed. Is this realistic? If this has been obtained with the CG models, or via the theoretical model, this should be proven and validated by means of atomistic simulations, to ensure that the Martini-based CG models are accurate in this sense. The solvation of Cit ions can be, for example, strongly different in atomistic and CG models. In this sense, the free energies obtained with CG models can be different from the atomistic ones as the entropy in the models is different.

The Reviewer is interpreting correctly the results from our theoretical model, and, as with point 2, this comment concerns the affinity between gold and citrate. To attend this comment, we note that, first, high level DFT calculations revealed an association energy of 40.9 kcal mol⁻¹ between citrate and gold surfaces, proving the covalent character of the two substrates' interaction (Al-Johani, H., et al. **2017**, *Nat. Chem.* DOI: 10.1038/nchem.2752). Second, the experimental characterization of citrate-capped gold nanoparticles revealed that the average deprotonation state (α) of the coating citrate molecules is $2 < \alpha < 3$ (Park, J. W., et al. **2014**, *J. Am. Chem. Soc.* DOI: 10.1021/ja4097384).

As commented by the Reviewer, our theoretical model predicts a citrate-gold binding energy of 20.0 kcal mol⁻¹ for a doubly deprotonated state ($\alpha = 2$), whereas for full deprotonation ($\alpha = 3$), the value increases to 53.1 kcal mol⁻¹. Interpolating these limiting values, we find that the DFT binding energy of 40.9 kcal mol⁻¹ is reached at $\alpha \sim 2.6$, in good agreement with estimates from experiments. Following the suggestions of the Reviewer, we have modified the first subsection of the 'Results and Discussion' to include the comparison made above, highlighting the consistency between our theoretical predictions and data from experiments and quantum mechanical simulations.

6. It is unclear how the theoretical model used by the authors treat ionic competitions and cooperative effects. For example, when the ionic strength increases in the system, don't the authors observe ionic clusters in solution and on the surface of NPs? I believe that this can have strong local effects that are poorly captured by average analyses and exceed simple additive approaches. This makes me wondering about the accuracy of the theoretical model used. Moreover, in CG models, where ions are typically modeled as CG beads representing one ion with its solvation shell, such local clustering effects could be impaired.

We refer the Reviewer to the response of point #1 and #2. In short, the formulation of the theoretical model was meant to predict the number of ligands chemisorbed on spherical, rigid nanoparticles. The covalent character of the citrate-gold interaction and the weak coupling between surface citrate molecules and sodium counterions support the negligible role that ion-pairing and ion-exchange play in the monolayer's composition.

7. Last, it seems to me that the take home message of this paper aims at being general. The authors clearly say in the final part of the conclusion that their results clarify how this work clarifies how the size of the NPs, their surface charge density and the ionic strength control the aggregation of the NP. In my opinion this is true only for the specific system studied herein and in particular for the conditions explored in the models. In fact, if the authors keep all these features constant, but increase or decrease the temperature in the system, the

ISTITUTO ITALIANO
DI TECNOLOGIA

results would be probably different. This is a key reason why I believe that, provided that the technical points mentioned above are clarified, this work in its current form provides an interesting study on a specific case, but I doubt about the general relevance of the study on a wider perspective.

As we explained in point #3, and now clarified throughout the manuscript, there are two components to our study. The first of these is a theoretical framework that predicts the number of bound ligands onto spherical nanoparticles. By evaluating our theoretical model on citrate, we determined the surface coverage for that particular ligand. The second aspect of our study is the construction of dispersion state phase diagrams that are extendible to generic charged nanoparticles. These phase diagrams describe the effect of the surface charge, ionic strength, and size on nanoparticle aggregation at ambient temperature and pressure conditions. Of course, changing any thermodynamic variable (e.g., temperature, pressure, density, chemical potential, etc.) would result in different diagrams, as true for any phase diagrams; however, we hope that this Reviewer would agree with us that studying the entire thermodynamic space would be unfeasible. The combination of our theoretical model and the constructed phase diagrams enabled us to discuss our findings in terms of a colloid that is of paramount importance to nanotechnology (i.e., citrate capped metal nanoparticles). Based on the Reviewer's observation, the transferability of our theoretical model and phase diagrams seems to have been inadvertently bypassed. Thus, we have extended our analysis in the 'Results and Discussion' section to better communicate the generality of our results.

8. Overall, I find the text and the whole paper a bit too technical for a high-impact journal targeting a broad readership, while this would be better suited for a more technical specific journal.

We understand that the original version of our manuscript was unclear. Again, we apologize for it, and we have done our very best now in order to clarify the text and overall story, avoiding unnecessary technicalities. We believe that the revised version of the manuscript conveys much better the ultimate significance of our findings as well as the transferability of our results. Overall, we think that the quality and clarity of the manuscript have greatly improved, and we thank the Reviewer for his/her constructive criticisms.

ISTITUTO ITALIANO
DI TECNOLOGIA

Reviewer 2:

This article aims to determine the surface coverage of citrates on a gold nanoparticle using a thermodynamic cycles that allows them to treat the effects of solvation on the acquisition of citrates through the free energies of solvation of the nanoparticles with and without citrates, and the free energy of the acquisition of citrates in vacuum. Their results are not without experimental or computational precedent in the literature, but the work adds to the on-going research aimed at resolving this important quantity.

They noted the work of Park et al in providing an estimate of the citrate charge density at 1.7 citrate molecules per nm² versus the work by Rostek and co-workers with reported a value of 3.1 molecules per nm². The computational work done here to resolve the correct density has precedent in the work of Chong et al, *J. Phys. Chem.* 2018, 122, 28393-28401 using a very different approach. That work favored the densities seen by Park et al, and less. They also appear to anticipate the values found here for coverage of the various protonated citrates. The results of the present work should be compared and contrasted accordingly.

In summary, this work is worthy of publication but only once it considers and contrasts its results with all of the literature precedent.

We thank Reviewer #2 for his/her comments. We have performed a more thorough literature review on implicit-citrate models, thus including seven new references that discuss the citrate chemisorption onto metallic surfaces. Thanks to the Reviewer's comments, we have also found that the results of our theoretical model are in good agreement with other three computational studies (Tavanti, F., et al. **2019**, *Int. J. Mol. Sci.* DOI: 10.3390/ijms20143539, Chong, G., et al. 2018, *J. Phys. Chem. C* DOI: 10.1021/acs.jpcc.8b09666, and Tavanti, F., et al. **2015**, *J. Phys. Chem. C* DOI: 10.1021/acs.jpcc.5b05796).

REVIEWERS' COMMENTS

Reviewer #1 (Remarks to the Author):

The authors carefully considered my comments, and I appreciate the efforts that they put in clarifying the approach that has been followed and the rationale behind this. I am still not 100% in agreement with some of the points that are assumed in the method, and some aspects are still unclear to me - for example, if the Cit-NP is indeed very strong and nearly irreversible, how this compare with a CIT residence time of 150-200 ns onto the NP surface, if I am not wrong, obtained from the atomistic simulations? Isn't this a discrepancy? If not, and this is my misunderstanding, then I would just suggest the authors to clarify this briefly in the revised manuscript. If yes, then the authors should briefly motivate the reason of this discrepancy. With this, I believe that the paper can be accepted.

ISTITUTO ITALIANO
DI TECNOLOGIA

Reply to the Reviewer

First, we thank the Reviewer for recognizing the improvements incorporated into the revised version of our manuscript. Below, we report in black our response to the point raised by the Reviewer, whose comments are in red.

Reviewer 1:

The authors carefully considered my comments, and I appreciate the efforts that they put in clarifying the approach that has been followed and the rationale behind this. I am still not 100% in agreement with some of the points that are assumed in the method, and some aspects are still unclear to me - for example, if the Cit-NP is indeed very strong and nearly irreversible, how this compare with a CIT residence time of 150-200 ns onto the NP surface, if I am not wrong, obtained from the atomistic simulations? Isn't this a discrepancy?

If not, and this is my misunderstanding, then I would just suggest the authors to clarify this briefly in the revised manuscript. If yes, then the authors should briefly motivate the reason of this discrepancy.

The point raised by the Reviewer results from a misunderstanding on what we calculated the residence times for, which is not citrate but sodium counterions. We apologize this point was unclear. As a matter of fact, we used atomistic molecular dynamics (MD) simulations to assess the residence time of the sodium counterions rather than that of citrate molecules, since the latter are actually fixed throughout the simulations – as previously discussed with this Reviewer. The purpose of such simulations with the sodium counterions is to cross-validate the qualitative behavior observed for the same ions in our coarse-grained simulations. We have thus clarified this point by adding Passage (1) in the Results section and Passage (2) in the Supplementary Discussion 3.

- (1) *In these simulations, we computed the time for which sodium counterions remained in the vicinity of the capped NPs. Indeed, in both our coarse-grained and atomistic simulations, longer residence times for sodium counterions were found as the surface charge of the NP increased (Supplementary Fig. 5).*
- (2) *It is important to note that the sodium ions' residence times computed from our coarse-grained models offer a qualitative description of the lability of the counterions present, rather than structural insights on their exact binding mode onto metallic NPs. Thus, in order to verify the results of our coarse-grained models, we performed MD simulations with atomistic representations of equivalent citrate-capped AuNPs in electrolytic solutions. These simulations enabled us to calculate, at a finer resolution, the residence time of the sodium counterions. Indeed, our atomistic simulations of citrate-capped AuNPs displayed the same trend, that is, as the charge density of the NP increased, the sodium ions became more affine for the NP's surface, thus bonding for longer times (Supplementary Fig. 5).*